# Melatonin Decreases Alveolar Bone Loss in Rats with Experimental Periodontitis and Osteoporosis: A Morphometric and Histopathologic Study

**DOI:** 10.3390/biomedicines12030684

**Published:** 2024-03-19

**Authors:** Suat Serhan Altıntepe Doğan, Hülya Toker, Ömer Fahrettin Göze

**Affiliations:** 1Department of Periodontology, Faculty of Dentistry, Afyonkarahisar Health Sciences University, 03030 Afyonkarahisar, Turkey; 2Department of Periodontology, Faculty of Gulhane Dentistry, Health Sciences University, 06018 Ankara, Turkey; tokerhulya@gmail.com; 3Department of Pathology, Faculty of Medicine, Atlas University, 34408 Istanbul, Turkey; fahrettin.goze@atlas.edu.tr

**Keywords:** alveolar bone loss, osteoporosis, periodontitis, melatonin, animal model

## Abstract

Background: Periodontitis and post-menopausal osteoporosis include common chronic bone disorders worldwide, with similar etiopathogenetic events. This study evaluated the effect of systemic melatonin administration on the alveolar bone destruction of periodontitis progression in an experimental periodontitis model in osteoporotic rats. Methods: Forty-four Wistar rats were randomly divided into six experimental groups: control (C; *n* = 6); osteoporosis (O; *n* = 6); ligated periodontitis (LP; *n* = 8); osteoporosis- and periodontitis-induced (O+LP; *n* = 8); osteoporosis- and periodontitis-induced through 30 mg/kg/day melatonin administration (ML30; *n* = 8); and osteoporosis- and periodontitis-induced through 50 mg/kg/day melatonin administration (ML50; *n* = 8). The rats underwent bilateraloophorectomy and were maintained for 4 months to induce osteoporosis. After 4 months, 4-0 silk ligatures were placed submarginally around the mandibular first molar of each rat to induce experimental periodontitis, and melatonin was administered in the ML30 and ML50 groups for 30 days. Changes in alveolar bone levels were clinically measured, and tissues were histopathologically examined. Results: Osteoclastic activity in the LP and O+LP groups was significantly higher than in the other groups (*p* < 0.05), but was similar in the C, O, and ML30 groups (*p* > 0.05). RANKL activity was the highest in the O+LP group, while melatonin decreased RANKL activity in the melatonin-administered groups (*p* < 0.05). Systemically administered melatonin significantly decreased alveolar bone loss in the ML30 and ML50 groups compared with that in the periodontitis groups (*p* < 0.05). Conclusions: Melatonin inhibited alveolar bone destruction by decreasing the RANKL expression and inflammatory cell infiltration and increased osteoblastic activity in a rat model with osteoporosis and periodontitis.

## 1. Introduction

Periodontitis is a chronic multifactorial inflammatory disease associated with dysbiotic plaque biofilms and characterized by the progressive destruction of the tooth-supporting apparatus [1]. The process of infection initiated by microbial plaque results in an imbalance between the periodontal defence and destruction mechanisms, subsequently destructing the periodontium [1,2]. Any disease or situation that can modify the host defence or immune system can alter the response to bacterial challenge, either in favour of the microbes or the host. Thus, the prevention of periodontal disease involves increasing the host’s defence or reducing bacterial growth [3,4].

Chronic inflammatory diseases, such as periodontitis, affect various host defence systems [5]. Reactive oxygen species (ROS) are some of the host’s defence mechanisms against bacteria. The disruption of the balance between oxidants and antioxidants results in oxidative stress that accelerates the inflammatory process [6]. Oxidative stress is an important factor that can disrupt bone metabolism and increase alveolar bone loss in periodontitis [5]. Age-related oxidative stress has been shown to be related with impaired osteoblastic function [7]. A continuous production of ROS in osteoblasts causes lipid peroxidation, protein damage, and DNA lesions, resulting in osteoblastic dysfunction and apoptosis.

Osteoporosis is a skeletal disorder that debilitates bones and increases fracture risk by decreasing bone density and increasing fragility due to altered bone strength, microarchitecture, and metabolism [8]. Both osteoporosis and periodontitis are chronic inflammatory prevalent diseases that cause changes in bone metabolism. Significant alterations in bone metabolism, including reduced osteoblastic activity and increased osteoclastic activity, have been reported in both diseases [9]. Depending on these changes, it was found that postmenopausal women with osteoporosis have more significant periodontal attachment loss than those with average bone mineral density [10].

Increased ROS and oxidative stress in periodontitis are involved in bone destruction [5,6,11]. Furthermore, low bone mineral density (BMD) has been observed in post-menopausal osteoporosis [5,12]. Cervellati et al. reported that in post-menopausal osteoporosis, a reduced BMD and increased bone resorption are associated with increased serum hydroperoxides [13]. Wu et al. demonstrated that advanced oxidation protein products, a novel marker of oxidation-mediated protein damage, in post-menopausal osteoporosis, are associated with a low BMD and increased bone changes [14]. Therefore, antioxidants that can neutralise ROS and decrease oxidative stress might be useful in treating these oxidative stress-related diseases.

One of the strongest known antioxidants [13], melatonin, is a hormone synthesised in the pineal gland [15]. It can be demonstrated that it has both chronobiotic and cytoprotective functions [16]. Melatonin is crucial in regulating the daily biorhythm, [14] immunity, and sleep [15]; it also has antioxidant [13] and anti-inflammatory [17] effects. Numerous studies indicate that melatonin promotes healthy aging and slows the aging process [16]. High doses of melatonin have anti-inflammatory effects by reducing free radicals and proinflammatory cytokines [16,18]. Melatonin acts directly as an antioxidant by scavenging free radicals and increasing its antioxidant activity to high levels [19]. It indirectly downregulates pro-oxidant enzyme synthesis and upregulates antioxidant enzymes [20].

Melatonin affects bone metabolism by suppressing the ligand-receptor activator nuclear factor kappa-B (RANKL) and upregulating the osteoprotegerin (OPG) expression [4,21,22]. Arabaci et al. recently revealed that melatonin inhibits osteoclastic activity by reducing the RANKL expression in the bone. The inhibition of osteoclastic activity was evident by the decrease in the C-terminal telopeptide of type I collagen, a marker of collagen degradation [23]. The role played by melatonin in alveolar bone loss inhibition has been demonstrated in various studies [3,23,24]. Additionally, melatonin was found to increase BMD in post-menopausal women with osteopenia [25]. Kotlarcyzk et al. reported that melatonin improves physical symptoms associated with perimenopause, possibly by restoring the balance in bone remodelling [26]. Studies have shown that melatonin is well tolerated and does not cause any known toxicity [16]. In all studies conducted with rats and mice, it has been stated that melatonin between 0.5 and 100 mg of calcium enhance bone remodelling by reducing bone resorption and that it positively affects the BMD without significant side effects [21,22,27,28,29,30,31,32,33].

Hence, it is crucial to modulate the host immune response to prevent the development and spread of periodontal destruction. The improvement of the antioxidant defence mechanism and reduction of ROS prevent further damage from oxidative stress. The benefits of antioxidants have been previously reported [3,4,24,34]. Contrarily, there are few studies on the role of antioxidants on bone metabolism in chronic inflammatory bone disorders, such as periodontitis and osteoporosis. Based on the favourable effects of melatonin on bone tissue, we hypothesised that the disrupted bone metabolism in post-menopausal osteoporosis might also be ameliorated with melatonin administration. This study aimed to evaluate the effect of melatonin on the progression of alveolar bone destruction and in a rat model with experimental osteoporosis and periodontitis.

## 2. Materials and Methods

The local ethics committee of Animal Studies (B.30.2.CUM.0.01.00.00-50/48/06.06.2018) approved the study. The study protocol and the manuscript were created according to the “NC3Rs ARRIVE Guidelines, Animal Research: Reporting of In Vivo Experiments” [35]. Forty-four female Wistar rats (weight: 230-250 g) were used. All rats were kept in an experimental animal laboratory and fed, according to previous studies [3,23]. The required sample size was 8 in the periodontitis groups with a statistical power of 85% [3,36,37].

The rats were randomly divided [3,24,37,38] into six study groups as follows:Non-ligated control (C, *n* = 6);Ligated periodontitis-induced (LP, *n* = 8);Osteoporosis-induced (O, *n* = 6);Osteoporosis- and periodontitis-induced (O+LP, *n* = 8);Osteoporosis- and periodontitis-induced through 30 mg/kg melatonin administration (ML30, *n* = 8);Osteoporosis- and periodontitis-induced through 50 mg/kg melatonin administration (ML50, *n* = 8).

To induce osteoporosis, 30 rats underwent bilateral oophorectomy under anaesthesia (ketamine, 30 mg/kg intramuscularly [IM] and rompun, 5 mg/kg IM) [8]. All operations were performed by the same veterinarian surgeon. After the operation, the rats were maintained for 4 months [3].

After four months, experimental periodontitis was induced using a ligation method. In total, 32 rats were anaesthetised (ketamine, 30 mg/kg/day IM and rompun, 5 mg/kg IM), and 4-0 silk ligatures (Dogsan Ilac Sanayi, Istanbul, Turkey) were placed submarginally around the lower right first molars [38,39]. All the ligation procedures were performed by an experienced operator (S.S.A.D.). All sutures were ensured for 1 month to form periodontitis. After osteoporosis and periodontitis were established, a melatonin solution was administered intraperitoneally (30 mg/kg/day and 50 mg/kg/day) between 10:00 and 11:00 in the morning, for 30 days. Melatonin was dissolved in ethanol and was administered as a 0.5 mL solution daily to each rat. The same amount of placebo solution (0.5 mL ethanol) was consumed by all rats without melatonin administrations [3]. After 30 days of melatonin administration, the rats were sacrificed with an anaesthetic overdose.

### 2.1. Measurement of Alveolar Bone Loss

After the sacrifice, the right mandibles of the rats were excised. The distance between the cementoenamel junction (CEJ) and the crest of the alveolar bone was measured and defined as the lost bone. To visualise the CEJ, the mandibles were stained with methylene blue and examined under a stereomicroscope (Carl Zeiss, Stemi 2000 and Axiovison 4.8, Jena, Germany) (12× magnification) and analysed using integrated digital image software (Clemex, Vision Image Analysis Software, Vers. 8, Brossard, QC, Canada). We measured the alveolar bone loss at three different sites on both the buccal and lingual regions of the first molar (mesio-buccal, mid-buccal, disto-buccal, mesio-lingual, mid-lingual, and disto-lingual), and a mean value was recorded. A single calibrated examiner performed all measurements (S.S.A.D.). For calibration, the examiner measured 20 different tooth samples three times, which were not included in this study. The results of the measurements showed r = 0.99 meaning 99% reproducibility. After achieving a 99% accuracy, the examiner measured the samples of the present study.

### 2.2. Histopathological Evaluation

All histologic analyses were performed by a single examiner (Ö.F.G.) who was also blinded to the identities of the samples [40]. All the mandibles were placed in formalin solution (10% neutral buffered formalin [41,42]. After fixation for 24 h, the samples were decalcified with ethylenediaminetetraacetic acid for approximately 2 months. The decalcification solution was changed two times a week, until decalcification was completed. After decalcification, all samples were dehydrated in ethanol and embedded in paraffin for a histological examination of the tissues [43]. Subsequently, 5 µm thick serial sections were prepared and stained with haematoxylin and eosin (H&E) or stained immunohistochemically for RANKL. The periodontal tissues in the mesial and distal parts of the mandibular first molar tooth were observed.

Osteoblasts are cubic cells located adjacent to the periodontal ligament lining the osteoid and alveolar bone, and these were counted under 400× magnification. Five sites (the mesial coronal, mesial middle, distal coronal, distal middle, and apical regions) were examined for osteoblasts. The total osteoblast count of these regions was recorded [37].

Inflammatory cell infiltration (ICI), including that of neutrophils, lymphocytes, eosinophils, and macrophages, was evaluated around the first molar under 400× magnification. ICI was evaluated using semi-quantitative scoring, as no visible ICI (0), slightly visible ICI (1), moderately visible ICI (2), and dense ICI (3) [44].

The alveolar bone adjacent to the periodontal ligament was examined for the osteoclast count. Giant multinuclear cells lining the lacunae in contact with the bone were counted as osteoclasts.

### 2.3. RANKL Immunohistochemistry

RANKL immunohistochemistry was performed to evaluate bone destruction, according to previous studies [37,42]. After the deparaffinisation and dehydration of the sections, antigen retrieval was performed using a 10 mM sodium citrate buffer (pH 6.0) for 2 h at 70 °C. Thereafter, endogenic peroxidise activity was quenched with 3% hydrogen peroxide. The samples were incubated with primary antibodies overnight (anti-RANKL antibody (Abcam plc, Trumpington, Cambridge, UK)) (1:100) after incubation with normal rabbit serum for 30 min. The sections were incubated with biotinylated immunoglobulin G for 30 min after washing five times with phosphate-buffered saline (PBS), followed by washing several times with PBS, and incubating with a streptavidin-horseradish peroxidase-conjugated reagent for 30 min. To evaluate immunoreactivity, the samples were incubated with 3,3′-Diaminobenzidine after washing with PBS (three washes, 5 min each). The sections were counterstained with haematoxylin and analysed using light microscopy.

Alveolar bone areas surrounding the roots of the first molars were examined for RANKL+ staining in the areas of the bone surrounding teeth. The percentage of the RANKL+ area to the examined area was calculated. The density of staining in the evaluated area was scored as 0, no staining; 1, light staining; 2, mild staining; and 3; dense staining.

### 2.4. Statistical Analysis

Statistical analyses were performed using the SPSS for Windows software package (version 20.0). For descriptive statistics, the mean ± standard deviation and median (minimum-maximum) were used for the quantitative variables. Normality was tested with the Kolmogorov-Smirnoff test. Since normal distributions were not provided, the Kruskal–Wallis H test was used to compare categories of the qualitative variable with more than two categories. To examine the pairwise differences resulting from significant differences between the six groups, the Mann–Whitney U test with Bonferroni correction was used. *p* < 0.05 was considered statistically significant. For a power analysis, an alpha of 0.05 was selected for the calculation.

## 3. Results

### 3.1. Morphometric Analyses

Experimental osteoporosis was successfully established, and no complications were observed. Experimental periodontitis was successfully achieved. The ligation caused periodontal destruction and alveolar bone loss around mandibular first molar teeth [3]. Also, the animals did not show any obvious signs of systemic illness throughout the study and completed the study.

Alveolar bone destruction was significantly higher in the LP group than those in the C, O, O+LP, ML30, and ML50 groups (*p* < 0.001; Table 1). The control group had the lowest mean alveolar bone destruction among all the groups. (*p* < 0.001). Significant differences were found between the C-O and C-LP, O-OP, and O-LP groups (*p* = 0.004, *p* = 0.003, *p* = 0.011, and *p* = 0.008, respectively) upon examining the binary groups. (Table 1; Figure 1). There was no statistically significant difference in the alveolar bone losses between the ML30 and C groups; 30 mg melatonin decreased the bone loss to the level seen in the control group (Table 1; Figure 1).

### 3.2. Histopathological Analyses

Figure 2 shows the histological slides. Osteoclasts were observed in Howship’s lacunae along the bone surface. It was found that there is a significant difference in the number of osteoclasts among the six groups (*p* < 0.001; Table 1). The LP group had the highest mean for osteoclast numbers, while the C group had the lowest. The binary analysis revealed significant differences between the C-ML30, C-ML50, O-ML30, and O-ML50 groups (*p* < 0.001, *p* = 0.001, *p* = 0.001, and *p* = 0.011, respectively; Table 1). Significantly lower osteoclast counts were observed in the C and O groups than in the LP and O+LP groups (*p* < 0.001; Figure 2; Table 1). The highest number of osteoclasts was observed in the O+LP group. Both doses of melatonin significant decreased the number of osteoclasts compared to that in the LP and O+LP groups (*p* < 0.05; Table 1). However, there was no significant difference between the ML30 and ML50 groups (*p* > 0.05; Table 1).

Osteoblast cell counts demonstrated that the C group had the lowest cell counts among the groups, and the difference was statistically significant (*p* < 0.001). Group ML30 had the highest mean value for the number of osteoblasts. Significant differences were found between the Control-ML30, Control-ML50, Osteoporosis-ML30, and Osteoporosis-ML50 groups (*p* < 0.001, *p* = 0.001, *p* = 0.001 and *p* = 0.011, respectively). Also, there was no significant difference in osteoblast cells between the ML30 and ML50 groups (*p* > 0.05; Figure 2; Table 1).

The highest ICI scores were observed in the O+LP groups, while the lowest was observed in the C group (*p* < 0.001). The C and O groups had similar ICI scores (*p* > 0.05; Table 1). Significant differences were found among the binary groups C-LP, C-O+LP, and O-O+LP (*p* = 0.012, *p* = 0.003, and *p* = 0.036, respectively; Table 1).

### 3.3. RANKL Immunohistochemistry

There was a significant difference found among the six groups in relation to RANKL scores (*p* < 0.001). The differences between the C-O+LP, O-O+LP, and O+LP-ML30 groups were found to be significantly different (*p* = 0.001, *p* = 0.005, and *p* = 0.002, respectively). The group with both O+LP had the highest average RANKL scores, while the control group had the lowest. No significant differences were observed in the RANKL expression between the ML30 and ML50 groups (*p* > 0.05) (Table 1; Figure 3).

## 4. Discussion

The present study is the first to evaluate the effect of melatonin on bone tissue in an experimental rat model of periodontitis and osteoporosis. Both 30 mg/kg and 50 mg/kg melatonin dosages significantly decreased the alveolar bone destruction of periodontitis progression and inhibited osteoclasts. Unlike natural disease progression, ligature-induced periodontal disease has a more acute course of reaction against plaque. Nonetheless, advantages such as low price, easy manipulation, availability, similar anatomic structure, and response to periodontal treatment make this the preferred and most studied method for experimentally induced periodontitis. Our results demonstrate that ligation caused significant alveolar bone destruction in the LP and LP+O groups, and both doses of melatonin significantly decreased this destruction. Among the various established methods for inducing osteoporosis, such as a low calcium diet [36], retinoic acid use [45], bilateral ovariectomy [46], and immobilisation [47], oophorectomy was used in the present study. A major disorder caused by post-menopausal hormonal changes is osteoporosis [48]. Osteoporosis progresses slowly, parallel to the deterioration in bone metabolism [49].

Numerous studies have found that osteoporosis exacerbates alveolar bone loss from periodontal destruction [50,51,52]. Systemic reduction in BMD has been shown to decrease mandibular bone density and increase alveolar bone loss [53]. Svedha et al. suggested that chronic periodontitis causes more severe periodontal destruction in post-menopausal women with low BMDs and considered periodontitis as a risk indicator for osteoporosis in post-menopausal women [54]. Recently, a systematic review by Goyal et al. revealed that women with severe periodontitis might have an increased risk of osteoporosis and should be examined for systemic bone health [9]. Ichimaru et al.’s findings also supported those of previous studies and revealed that the systemic treatment of an osteoporosis-increased BMD prevents alveolar bone loss caused by bacterial lipopolysaccharides [55]. Amedei et al. [52] showed that the concurrent oestrogen deficiency and periodontitis led to increased alveolar bone damage in rats. Furthermore, pro-inflammatory cytokines, such as RANKL and interleukins, were found elevated with increases in the phosphate level, and these increased levels were evaluated as an element of the alveolar bone damage mechanism in periodontitis. Conversely, Weyant et al. did not find any correlation between systemic BMD and the clinical manifestations of periodontal tissue destruction [56]. Supporting this, Anbinder et al. suggested that osteoporosis did not exacerbate bone destruction caused by periodontitis [57]. In the present study, osteoporosis along with ligature placement increased alveolar bone loss more severely than that in the periodontitis-only group, but there is no statistically significant difference between LP and LP+O groups. However, osteoporosis without ligature placement did not affect alveolar bone loss. Thus, we determined that osteoporosis alone did not cause significant destruction of the alveolar bone.

Menopause involves the cessation of ovarian follicular activity and menstrual cycles, with various clinical consequences, such as hormonal alterations, sleep disorders, vasomotor symptoms, and osteopenia/osteoporosis [58]. Aging and menopause together reduce melatonin levels [59]. Recently, Tresguerres et al. revealed that a melatonin-rich diet improved the decrease in bone trabecular numbers, bone volume, and thickness caused by aging in rats [60]. Amstrup et al. stated that melatonin increases the trabecular thickness in the distal tibia and BMD at the femoral neck and lumbar spinal cord in a dose-dependent manner in women with osteopenia after menopause [25]. In addition, daily melatonin intake reportedly increases bone health and related quality of life in perimenopausal women [26]. These studies evaluated the relationship between melatonin’s chronobiotic effect, its intake hours, and its cytoprotective and anti-aging effects. In studies focusing on the chronobiotic effect, lower doses of melatonin were found to be effective. However, the impact of melatonin on bone occurs through a distinct pathway from that of the receptors involved in its chronobiotic effect [16,22,61]. It has been observed that higher doses of melatonin are required to have an impact on bone remodelling [62]. Studies have shown that doses of melatonin up to 100 mg are non-toxic in humans [63,64]. In a previous study on rats, melatonin administration decreased interleukin (IL)-1β, tumour necrosis factor (TNF)-α, and malondialdehyde in periodontal tissues, decreased alveolar bone loss, and increased glutathione peroxidase levels [34]. The effect of melatonin on periodontal destruction might be because of decreased osteoclastic activity, collagen degradation, and inflammation, as reported by Arabaci et al. [23]. Recently, Balci Yuce et al. and Sarıtekin et al. also stated that the systemic consummation of 10 mg/kg of melatonin decreases alveolar bone loss, periodontal inflammation, and tartrate-resistant acid phosphatase-positive osteoclasts, while it had no effect on inducible nitric oxide synthase and apoptotic proteins [3,65]. Halici et al. reported that administering 30 mg/kg of melatonin intraperitoneally may regulate osteoclastic activity and stimulate osteoblast differentiation and matrix mineralisation [66]. Koyama et al. showed that, intraperitoneally, a 50 mg/kg dose of melatonin hinders the osteoclasts and increases the efficiency of the osteoblasts [22]. In our study, we used melatonin dosages of 30 and 50 mg/kg, and melatonin reduced osteoclasts and ICI in osteoporotic rats. Previous research has examined the effects of a 10 mg/kg dose of melatonin and found that it can reduce bone loss in the presence of type 2 diabetes and periodontitis [3,23,67]. Melatonin doses of 30 and 50 mg/kg were administered to rats with experimental periodontitis and osteoporosis for the first time in this animal study. It was determined that increasing the melatonin dose to 50 mg/kg did not yield positive results. Although there was a rise in the count of osteoblasts in melatonin-administered groups, the difference was not statistically significant.

Oestrogen deficiency causes stromal and osteoblastic cells to become more susceptible to cytokines, such as RANKL, IL-1, and IL-6 [68]. RANK, RANKL, and OPG are regulatory proteins crucial to osteoclastogenesis. RANKL binding to RANK increases the lifespan, proliferation, differentiation, and activity of osteoclasts [69]. Active T-cells in periodontal tissues may cause alveolar bone destruction directly by upregulating RANKL, or indirectly by increasing RANKL synthesis in osteoblasts through cytokines, such as TNF-α, IL-11, and IL-17 [70]. RANKL expression increases with both osteoporosis and periodontitis, and RANKL has also been suggested to be a determining element in the early diagnosis of osteoporosis [50]. Luo et al. reported that bilateral ovariectomy in rats increases RANKL, OPG, and IL-6 levels in periodontal tissues and alters bone metabolism [51]. Increased pro-inflammatory cytokines in the periodontal microenvironment increase the quantity of osteoclast precursors by differentiating them into mature osteoclasts and prolonging the lifespan of these osteoclasts [71]. Similarly, among the present study groups, a significant difference was noted in terms of RANKL activity. The LP and O+LP groups had high RANKL activity, whereas the control group had low RANKL activity. Furthermore, groups administered with 30 mg/kg and 50 mg/kg doses of melatonin exhibited similar RANKL activity.

This study had several limitations. First, there have been no clinical studies using melatonin as an adjunctive treatment for periodontitis and osteoporosis. In addition, the precise mechanisms by which melatonin purifies oxidants and prevents osteoporotic complications are not fully understood. Hence, the outcomes attained could not be compared with the results of other studies. Second, we evaluated alveolar bone changes via stereomicroscopy, which provides a two-dimensional evaluation. A three-dimensional analysis, such as micro-computed tomography would be more precise in determining the CEJ and alveolar bone distance. Third, gene activities and protein levels indicating bone formation and bone loss were not determined. Fourth, controls ML30 and ML50 in animals with periodontitis but without osteoporosis were not included in this study and were not evaluated in terms of periodontitis alone. Fifth, another limitation was the absence of biochemical analyses like Western blot or PCR. Lastly, in this study, the therapeutic effect of melatonin in periodontitis was assessed, but its preventive effect was not evaluated. It is important to evaluate the effectiveness of melatonin at both lower and higher doses.

## 5. Conclusions

Based on the limitations of this study, melatonin could be a potential host modulatory agent for periodontitis management. Within the limitations of this study, the systemic administration of melatonin has been successfully used for decreasing bone destruction in a rat model of osteoporosis and periodontitis by inhibiting both osteoclasts and the RANKL expression. However, further studies with advances in analysis such as microcomputed tomography are needed to investigate the mechanisms of action for the effects of melatonin in periodontitis progression concurrent with osteoporosis. Further animal and clinical studies are necessary to determine the minimum effective dose of melatonin as an adjuvant agent for treating periodontitis and osteoporosis in humans.

## Figures and Tables

**Figure 1 biomedicines-12-00684-f001:**
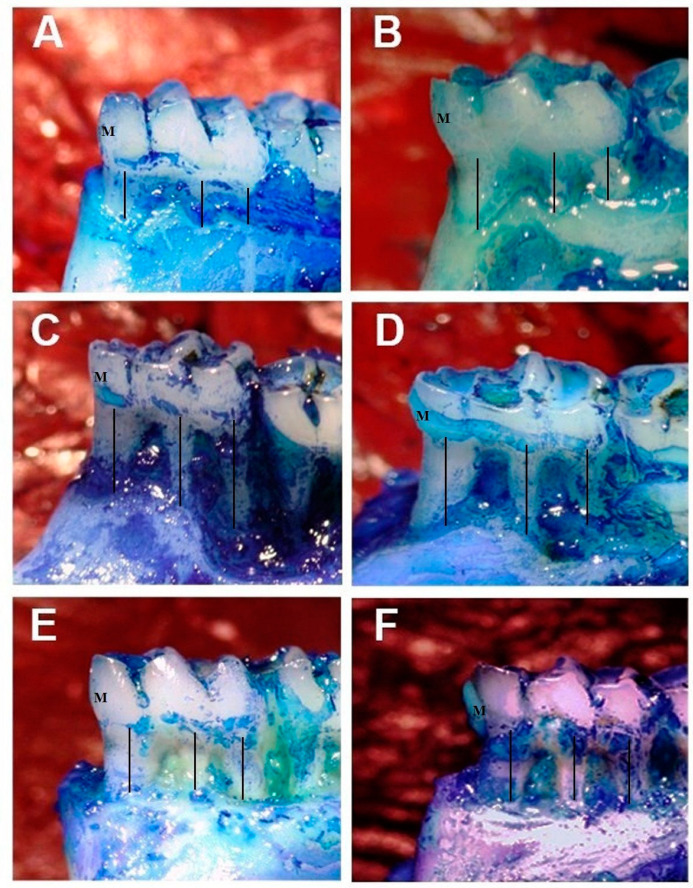
Representative stereomicroscopy images of the groups (12× magnification) (Distance between the cementoenamel junction and alveolar bone crest = Black lines, Mesial side of first molar = M). (**A**): Control group (C, *n* = 6), (**B**): Osteoporosis group (O, *n* = 6), (**C**): Periodontitis group (LP, *n* = 8), (**D**): Osteoporosis and periodontitis group (O+LP, *n* = 8), (**E**): 30 mg/kg melatonin-administered group (ML30, *n* = 8), (**F**): 50 mg/kg melatonin-administered group (ML50, *n* = 8).

**Figure 2 biomedicines-12-00684-f002:**
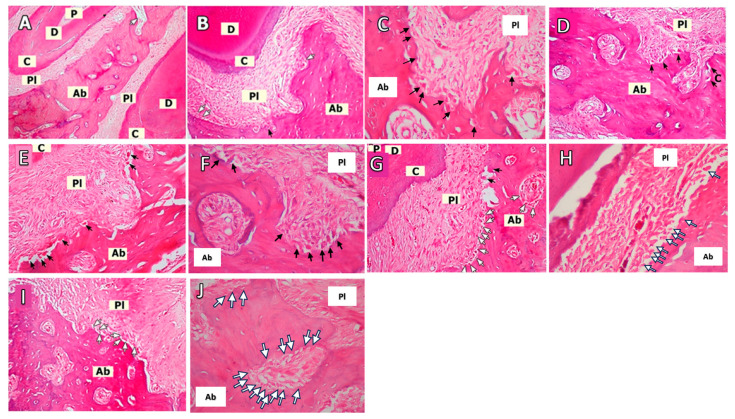
Representative histological images of the haematoxylin-eosin staining in the groups (Osteoclastic activity = Black arrows, Osteoblasts = White arrows). (**A**): Control group (C, *n* = 6) (100× magnification), (**B**): Osteoporosis group (O, *n* = 6) (200× magnification), (**C**): Osteoporosis group (400×) magnification, (**D**): Ligature-induced periodontitis group (LP, *n* = 8) (200× magnification), (**E**): Osteoporosis and periodontitis group (O+LP, *n* = 8) (200× magnification), (**F**): Osteoporosis and periodontitis group (400× magnification), (**G**): 30 mg/kg melatonin-administered group (ML30, *n* = 8) (200× magnification), (**H**): 30 mg/kg melatonin-administered group (400× magnification), (**I**): 50 mg/kg melatonin-administered group (ML50, *n* = 8) (200× magnification), (**J**): 50 mg/kg melatonin-administered group (400× magnification). Ab: alveolar bone, C: cement, D: dentin, Pl: periodontal ligament, P: pulp.

**Figure 3 biomedicines-12-00684-f003:**
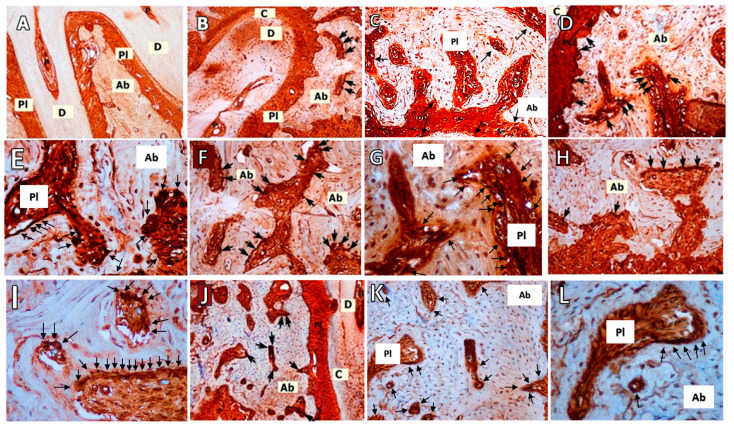
Representative images of RANKL immunohistochemical staining. Black arrows signify cells positive for RANKL immunohistochemistry. The control group revealed no RANKL-positive cells. (**A**): Control group (C, *n* = 6) (100× magnification), (**B**): Osteoporosis group (O, *n* = 6) (100× magnification), (**C**): Osteoporosis group (O, *n* = 6) (200× magnification), (**D**):Ligature-induced periodontitis group (LP, *n* = 8) (200× magnification), (**E**): Ligature-induced periodontitis group (LP, *n* = 8) (400× magnification), (**F**): Osteoporosis and periodontitis group (O+LP, *n* = 8) (200× magnification), (**G**): Osteoporosis and periodontitis group (O+LP, *n* = 8) (400× magnification), (**H**): 30 mg/kg melatonin-administered group (ML30, *n* = 8) (200× magnification), (**I**): 30 mg/kg melatonin-administered group (ML30, *n* = 8) (400× magnification), (**J**): 50 mg/kg melatonin-administered group (ML50, *n* = 8) (100× magnification). (**K**): 50 mg/kg melatonin-administered group (ML50, *n* = 8) (200× magnification). (**L**): 50 mg/kg melatonin-administered group (ML50, *n* = 8) (400× magnification). Ab: alveolar bone, C: cement, D: dentin, Pl: periodontal ligament.

**Table 1 biomedicines-12-00684-t001:** Comparing study groups for data analysis.

Variables		C Group	O Group	LP Group	O+LP Group	ML30 Group	ML50 Group	*p* Value
Mean alveolar bone loss	Mean ± SD	0.58 ± 0.16	0.65 ± 0.19	1.21 ± 0.23	1.16 ± 0.17	0.86 ± 0.18	0.96 ± 0.19	<0.001 ^a^
	Median (Min-Max)	0.59 (0.33–0.75)	0.57 (0.49–1.02)	1.12 (0.92–1.54)	1.19 (0.83–1.32)	0.91 (0.53–1.04)	0.98 (0.57–1.14)	
Osteoclast numbers	Mean ± SD	1.17 ± 1.17	4.83 ± 1.94	20.00 ± 12.40	28.00 ± 14.98	10.13 ± 3.83	12.38 ± 3.74	<0.001 ^a^
	Median (Min-Max)	1.0 (0.0–3.0)	4.5 (3.0–8.0)	16.5 (5.0–40.0)	26.5 (9.0–50.0)	9.0 (6.0–16.0)	11.50 (8.0–19.0)	
Osteoblast numbers	Mean ± SD	20.67 ± 9.07	34.50 ± 10.31	69.38 ± 10.46	64.13 ± 9.14	93.75 ± 16.68	82.88 ± 14.18	<0.001 ^a^
	Median (Min-Max)	17.5 (12.0–35.0)	33.5 (20.0–48.0)	69.0 (54.0–85.0)	65.5 (48.0–76.0)	91.0 (74.0–120.0)	83.0 (64.0–108.0)	
RANKL scores	Mean ± SD	0.67 ± 0.52	0.83 ± 0.75	2.13 ± 0.99	2.88 ± 0.35	0.88 ± 0.64	1.25 ± 0.46	<0.001 ^a^
	Median (Min-Max)	1.0 (0.0–1.0)	1.0 (0.0–2.0)	2.5 (1.0–3.0)	3.0 (2.0–3.0)	1.0 (0.0–2.0)	1.0 (1.0–2.0)	
Inflammatory cell infiltration scores	Mean ± SD	0.67 ± 0.52	1.0 ± 0.0	2.0 ± 0.76	2.13 ± 0.64	1.13 ± 0.35	1.50 ± 0.53	<0.001 ^a^
	Median (Min-Max)	1.0 (0.0–1.0)	1.0 (1.0–1.0)	2.00 (1.0–3.0)	2.0 (1.0–3.0)	1.0 (1.0–2.0)	1.5 (1.0–2.0)	

SD: Standard Deviation, Min: Minimum, Max: Maximum, ^a^: Kruskal–Wallis H test.

## Data Availability

The data that support the findings of this study are available from the corresponding author upon reasonable request.

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
