# Peer review of "Melatonin Decreases Alveolar Bone Loss in Rats with Experimental Periodontitis and Osteoporosis: A Morphometric and Histopathologic Study"

_biomedicines, 2024, doi:10.3390/biomedicines12030684_

Round 1

Reviewer 1 Report

Comments and Suggestions for Authors

Dear author, the article is very interesting, and the topic is very modern in oral and maxillofacial surgery.

I am honored to share with you just few suggestions to try to improve the paper.

The present study is well designed and conducted, and the manuscript is quite clear.

The English is good and spelled correctly but to be published it needs of some adjustment.

There are some comments below.

Abstract: The abstract correctly summarizes the study design and purpose as the title as well.

Keywords: The keywords are correct and perfectly fitting the study design, I would add “animal model” if it is possible.

Introduction:

You should improve your references adding also concepts and sentence to explain why this bone resorption is important in osteoporosis. The process of Bone loss after tooth extraction is significant in the first six months post-extraction yes, but especially in the fist weeks of healing due to the granulation tissue and the contractile action of Myofibroblast how it was demonstrated in the in vitro studies such as the following one that you could mention. In this animal study, authors used micro-CBCT instead of stereomicroscopy.

Covani U, Giammarinaro E, Panetta D, et all. Alveolar Bone Remodeling with or without Collagen Filling of the Extraction Socket: A High-Resolution X-ray Tomography Animal Study. J Clin Med. 2022 Apr 29;11(9):2493. doi: 10.3390/jcm11092493.

- Materials and methods

Please cite articles with similiar structure to justify the use of rats model such as the following animal study:

Nocini PF, Menchini Fabris GB, Gelpi F, Lotti J, Favero V, Zanotti G, Jurlaro A, Rosskopf I, Lotti T, Barone A, Castegnaro G, De Santis D. Treatment of skin defects with growth factors and biodegradable collagen carrier: histological evaluation in animal model. J Biol Regul Homeost Agents. 2017 Apr-Jun;31(2 Suppl. 2):1-13.

-        Discussion

Moreover, you should report also other clinical articles contrasting what you have reported.

Regarding the fact that it is very important to be atraumatic and minimally invasive as possible in all treatment in osteoporosis

Anyway, the rest of article and conclusion are well done.

Please explain more about the limitation of this study in the discussion, also about the in animal nature of the study because admitting the limitation is a good aspect of a single study bringing researcher to improve for future in vitro or clinical study.

I hope these suggestions may help you to publish the article.

Best regard,

Author Response

Dear reviewer, thank you for your valuable feedback and constructive criticism. Your insights have helped us improve the quality of our article, and we greatly appreciate your efforts. 

"The keywords are correct and perfectly fitting the study design; I would add "animal model" if it is possible."

The term "animal model" has been included in the keywords section of the study.

You should improve your references adding also concepts and sentence to explain why this bone resorption is important in osteoporosis. The process of Bone loss after tooth extraction is significant in the first six months post-extraction yes, but especially in the fist weeks of healing due to the granulation tissue and the contractile action of Myofibroblast how it was demonstrated in the in vitro studies such as the following one that you could mention. In this animal study, authors used micro-CBCT instead of stereomicroscopy.

Covani U, Giammarinaro E, Panetta D, et all. Alveolar Bone Remodeling with or without Collagen Filling of the Extraction Socket: A High-Resolution X-ray Tomography Animal Study. J Clin Med. 2022 Apr 29;11(9):2493. doi: 10.3390/jcm11092493.

Dear reviewer, since this was a periodontitis study, we analyzed the mentioned reference. Although it was not directly related to periodontitis, we still used it as the no 2 reference to discuss the process of bone destruction and repair in the periodontium during periodontitis.

Please cite articles with similiar structure to justify the use of rats model such as the following animal study:

Nocini PF, Menchini Fabris GB, Gelpi F, Lotti J, Favero V, Zanotti G, Jurlaro A, Rosskopf I, Lotti T, Barone A, Castegnaro G, De Santis D. Treatment of skin defects with growth factors and biodegradable collagen carrier: histological evaluation in animal model. J Biol Regul Homeost Agents. 2017 Apr-Jun;31(2 Suppl. 2):1-13.

We have added the reference number mentioned earlier to the text as 23.

Discussion

Moreover, you should report also other clinical articles contrasting what you have reported.

In this study, the effectiveness of melatonin was tested for the first time in rats with periodontitis and osteoporosis. However, there are no similar studies with which we can compare our results in detail. We made an effort to reinforce our findings with additional studies that are similar in nature.

Menopause involves the cessation of ovarian follicular activity and menstrual cycle, with various clinical consequences, such as hormonal alterations, sleep disorders, vasomotor symptoms, and osteopenia/osteoporosis.48 Aging and menopause together reduce melatonin levels.49 Recently, Tresguerres et al. revealed that a melatonin-rich diet improved the decrease in bone trabecular numbers, bone volume, and thickness caused by aging in rats.50 Amstrup et al. stated that melatonin increased the trabecular thickness in the distal tibia and BMD at the femoral neck and lumbar spinal cord in a dose-dependent manner in women with osteopenia after menopause.20 In addition, daily melatonin intake reportedly increased bone health and related quality of life in perimenopausal women.21 Melatonin was also shown to increase the regenerative capacity of bone defects in rabbits.51 In a previous study in rats, melatonin administration decreased interleukin (IL)-1β, tumour necrosis factor (TNF)-α, and malondialdehyde in periodontal tissues, decreased alveolar bone loss, and increased glutathione peroxidase levels.22 The effect of melatonin on periodontal destruction might be because of decreased osteoclastic activity, collagen degradation, and inflammation, as reported by Arabaci et al.18 Recently, Balci Yuce et al. also stated that systemic consummation of 10 mg/kg melatonin decreased alveolar bone loss, periodontal inflammation, and tartrate-resistant acid phosphatase-positive osteoclasts, while it had no effect on inducible nitric oxide synthase and apoptotic proteins.3 Halici et al. reported that administering 30 mg/kg of melatonin intraperitoneally may regulate osteoclastic activity and stimulate osteoblast differentiation and matrix mineralisation.52 Koyama et al. showed that intraperitoneally a 50 mg/kg dose of melatonin hindered the osteoclasts and increased the efficiency of the osteoblasts.17 In our study, we used melatonin dosages 30 and 50 mg/kg and melatonin reduced osteoclasts and ICI in osteoporotic rats. Previous research has examined the effects of a 10mg/kg dose of melatonin and found that it can reduce bone loss in the presence of type 2 diabetes and periodontitis3,18,53 Melatonin doses of 30 and 50 mg/kg were administered to rats with experimental periodontitis and osteoporosis for the first time in this animal study. It was determined that increasing the melatonin dose to 50 mg/kg did not yield positive results. Although there was a rise in the count of osteoblasts in melatonin-administered groups, the difference was not statistically significant.

Regarding the fact that it is very important to be atraumatic and minimally invasive as possible in all treatment in osteoporosis.

We have taken note of your opinion and have implemented all necessary edits and references to ensure the accuracy of the article. Thank you for bringing this to our attention.

Please explain more about the limitation of this study in the discussion, also about the in animal nature of the study because admitting the limitation is a good aspect of a single study bringing researcher to improve for future in vitro or clinical study.

The study limitations were thoroughly discussed, and necessary corrections and additions were highlighted in yellow.

Reviewer 2 Report

Comments and Suggestions for Authors

The authors studied the effect of melatonin on alveolar bone loss in rats with osteoporosis and/or experimental periodontitis. They showed that melatonin (30-50 mg/kg) inhibited periodontal decay in the melatonin treated osteoporosis/periodontitis animals. Essential controls are missing, although they are mentioned as limitations of this study, so the authors were well aware of this omission in their experimental design (periodontitis without osteoporosis, periodontitis treated with melatonin, osteoporosis treated with melatonin). The authors has to add these experiments as they are essential in interpreting the results. Furthermore, a power calculation was added, but were the multiple groups tested taken into account or was it just a comparison between a test group and a control.

With regard to the tables and figures, are the SD or SE depicted? Furthermore, was there a check of the effects of multiple testing?

Where the data normal on non-normal distributed? It is tested, but it is not mentioned in the results section.

Finally, a dose of 30-50 melatonin / kg body weight is very high for human. What will be the effect of clinical acceptable doses.

Author Response

The authors studied the effect of melatonin on alveolar bone loss in rats with osteoporosis and/or experimental periodontitis. They showed that melatonin (30-50 mg/kg) inhibited periodontal decay in the melatonin treated osteoporosis/periodontitis animals. Essential controls are missing, although they are mentioned as limitations of this study, so the authors were well aware of this omission in their experimental design (periodontitis without osteoporosis, periodontitis treated with melatonin, osteoporosis treated with melatonin). The authors has to add these experiments as they are essential in interpreting the results. Furthermore, a power calculation was added, but were the multiple groups tested taken into account or was it just a comparison between a test group and a control.

With regard to the tables and figures, are the SD or SE depicted? Furthermore, was there a check of the effects of multiple testing?

Where the data normal on non-normal distributed? It is tested, but it is not mentioned in the results section.

Dear Reviewer, we appreciate your careful criticism, which has highlighted the statistical shortcomings of our article. We are grateful for this opportunity to improve our work. A professional statistician Dr. Batuhan Bakırarar (PhD), has reviewed all the statistics in the article, and necessary corrections and new analyses have been made to address the issues you raised. Thank you again for your valuable feedback.

Dr. Batuhan Bakırarar (PhD)

https://orcid.org/0000-0002-5662-8193

Finally, a dose of 30-50 melatonin / kg body weight is very high for human. What will be the effect of clinical acceptable doses.

Dear reviewer, We would like to clarify that our study used a rat animal model and tested various doses of melatonin. However, we did not test the dose of 50 mg/kg. Furthermore, to the best of our knowledge, no other study has created and tested a study model with both osteoporosis and periodontitis. According to recent studies, the ideal dosage for human use is 10 mg/kg.1–6 However, these studies did not include models for osteoporosis and periodontitis. Therefore, future studies, both clinical and animal-based, are necessary to determine the minimum safe dosage for human use. These studies will provide more precise information on the subject.

Thank you for taking the time to review our work.

Best regards.

References

  1. Tresguerres IF, Tamimi F, Eimar H, et al. Melatonin Dietary Supplement as an Anti-Aging Therapy for Age-Related Bone Loss. Rejuvenation Res. 2014;17(4):341-346. doi:10.1089/rej.2013.1542
  2. Halıcı M, Öner M, Güney A, Canöz Ö, Narin F, Halıcı C. Melatonin promotes fracture healing in the rat model. Eklem Hastalik Cerrahisi. 2010;21(3):172-177.
  3. Kose O, Arabaci T, Kara A, et al. Effects of Melatonin on Oxidative Stress Index and Alveolar Bone Loss in Diabetic Rats With Periodontitis. J Periodontol. 2016;87(5):e82-e90. doi:10.1902/jop.2016.150541
  4. Amstrup AK, Sikjaer T, Heickendorff L, Mosekilde L, Rejnmark L. Melatonin improves bone mineral density at the femoral neck in postmenopausal women with osteopenia: a randomized controlled trial. J Pineal Res. 2015;59(2):221-229. doi:10.1111/jpi.12252
  5. Balci Yuce H, Karatas O, Aydemir Turkal H, et al. The Effect of Melatonin on Bone Loss, Diabetic Control, and Apoptosis in Rats With Diabetes With Ligature-Induced Periodontitis. J Periodontol. 2016;87(4):e35-43. doi:10.1902/jop.2015.150315
  6. Sarıtekin E, Üreyen Kaya B, Aşcı H, Özmen Ö. Anti‐inflammatory and antiresorptive functions of melatonin on experimentally induced periapical lesions. Int Endod J. 2019;52(10):1466-1478. doi:10.1111/iej.13138

Reviewer 3 Report

Comments and Suggestions for Authors

The study is interesting however several points need to be addressed. 

-The study lacks information on sample size calculation.

- The method of randomization and allocation is not explained in the methods.

- The method of ligation needs to be better described in the manuscript, as the measurement was only taken for the first molar and not the other teeth.

- It is not clear whether the resorption and osteoclast activity is due to osteoporosis or periodontitis. This should be clarified, as the changes with the melatonin are visible histologically.

- The clinical relevance should be emphasized in the discussion.

- Lastly, there is a typo on page 3 that needs to be corrected - "aesthetic overdose". 

Comments on the Quality of English Language

The authors need to proofread and edit their work for English language errors, as well as correct any typos that may be present.

Author Response

Dear reviewer, thank you for your valuable feedback and constructive criticism. Your insights have helped us improve the quality of our article, and we greatly appreciate your efforts.

Best regards.

The study lacks information on sample size calculation.

 The sample size for the study was determined by taking into account previous similar studies. 1–7  The minimum number of rats required for statistical evaluation was prioritized, and the number of 8 rats used in similar studies was determined while also considering ethical values regarding animal studies. However, since no action was taken in the control group and the researchers had conducted many studies on this subject before, the number of O groups was reduced to 6. 1–7

- The method of randomization and allocation is not explained in the methods

As in similar experimental rat studies, the characteristics of the rats were specified, and all were accepted as standard. In this way, groups were formed randomly. Necessary corrections were made and added to the text

The method of ligation needs to be better described in the manuscript, as the measurement was only taken for the first molar and not the other teeth.

The relevant section was reviewed and necessary additions were made based on the previous studies.

 It is not clear whether the resorption and osteoclast activity is due to osteoporosis or periodontitis. This should be clarified, as the changes with the melatonin are visible histologically.

Evaluations regarding your criticism have been made. Histopathological analyses have been made by an experienced pathologist, F.G.(Prof. Dr. Ph.D.), and our histological sections have been scrutinized under a stereomicroscope. Additionally, immunohistochemical methods have supported our histological section results.

The clinical relevance should be emphasized in the discussion.

The melatonin doses used in this study were selected based on previous research. However, further animal and clinical studies are needed to support our findings. Our study is the first in its field, and further research is necessary to validate the clinical significance of our findings. The limitations section of the study restates this situation.

 Lastly, there is a typo on page 3 that needs to be corrected - "aesthetic overdose". 

The entire article was re-evaluated for academic English editing by reviewing spelling errors. The relevant academic English editing certificate has been added.

Thank you for reading our manuscript and sharing your valuable opinions.

  1. Balci Yuce H, Toker H, Yildirim A, Tekin MB, Gevrek F, Altunbas N. The effect of luteolin in prevention of periodontal disease in Wistar rats. J Periodontol. 2019;90(12):1481-1489. doi:10.1002/JPER.18-0584
  2. Balci Yuce H, Toker H, Goze F. The histopathological and morphometric investigation of the effects of systemically administered boric acid on alveolar bone loss in ligature-induced periodontitis in diabetic rats. Acta Odontol Scand. 2014;72(8):729-736. doi:10.3109/00016357.2014.898789
  3. Toker H, Ozdemir H, Balci Yuce H, Goze F. The effect of boron on alveolar bone loss in osteoporotic rats. J Dent Sci. 2016;11(3):331-337. doi:10.1016/j.jds.2016.03.011
  4. Balci Yuce H, Toker H, Yildirim A, Tekin MB, Gevrek F, Altunbas N. The effect of luteolin in prevention of periodontal disease in Wistar rats. J Periodontol. 2019;90(12):1481-1489. doi:10.1002/JPER.18-0584
  5. Karatas O, Balci Yuce H, Taskan MM, et al. The effect of vanillic acid on ligature-induced periodontal disease in Wistar rats. Arch Oral Biol. 2019;103:1-7. doi:10.1016/j.archoralbio.2019.05.010
  6. Çalışır M, Akpınar A, Poyraz Ö, Göze F, Çınar Z. Humic Acid, a Polyphenolic Substance, Decreases Alveolar Bone Loss in Experimental Periodontitis in Rats. J Vet Dent. 2019;36(4):257-265. doi:10.1177/0898756420910531
  7. Karakan NC, Akpınar A, Göze F, Poyraz Ö. Investigating the Effects of Systemically Administered Strontium Ranelate on Alveolar Bone Loss Histomorphometrically and Histopathologically on Experimental Periodontitis in Rats. J Periodontol. 2017;88(2):e24-e31. doi:10.1902/jop.2016.160227

Round 2

Reviewer 2 Report

Comments and Suggestions for Authors

While the study has improved, there a still a number of major concerns which has to be addressed adequately:

·         When reporting medians, please add the Q1 and Q3 values, this is more insightful than the min – max values.

·         In table 1, please use . instead of ,

·         In table 1, it has no sense to report the RANKL and Infiltrate scores with two decimal points as the scores wee 1, 2 ,3 etc

·         In table 1, please give the mean ± SD or median IQR values as appropriate (also when writing the results). The data are either normal distributed or non-normal distributed, not both.

·         Figure 1 and figure 4, report the mean values or median values depending on the distribution of the data. This should match table 1. In fact it is redundant to report the same data in table 1 and in the figures.

·         It is unclear what is reflected in figure 4 and 5,  the data on the x and y axis have the same text.

·         Mention in the introduction and discussion that this is an animal study in which doses were used which are not useful in the clinic. It has to be assessed whether similar effects occur at clinically acceptable doses in human.

Author Response

Dear reviewer, I appreciate your valuable and meticulous observations. Thank you.

  • In table 1, please use . instead of ,

I want to thank you for your correction. I take full responsibility for the mistake and I appreciate you bringing it to my attention.

  • In table 1, it has no sense to report the RANKL and Infiltrate scores with two decimal points as the scores wee 1, 2 ,3 etc

We have received your request for this correction.

  • In table 1, please give the mean ± SD or median IQR values as appropriate (also when writing the results). The data are either normal distributed or non-normal distributed, not both.

As the data didn't follow a normal distribution, the Kruskal-Wallis H test was utilized. The p value and the test used are stated in a footnote under the table. Using minimum and maximum instead of IQR provides more comprehensive information about the data. Presenting the mean, standard deviation, median, minimum, and maximum values in a table provides a comprehensive overview of the data. Commenting that the median was higher than the other group based on the median data interpretation would be incorrect. Reducing data crowding and increasing readability can be achieved by avoiding writing each group's median, minimum, and maximum values. It was deemed more appropriate to present the information in the table.

  • Figure 1 and figure 4, report the mean values or median values depending on the distribution of the data. This should match table 1. In fact it is redundant to report the same data in table 1 and in the figures.

The corrections that you mentioned earlier have been implemented.

  • It is unclear what is reflected in figure 4 and 5, the data on the x and y axis have the same text.

After taking into account your criticism, we have decided to remove these graphics.

  • Mention in the introduction and discussion that this is an animal study in which doses were used which are not useful in the clinic. It has to be assessed whether similar effects occur at clinically acceptable doses in human.

The necessary corrections were made, marked in yellow, and shown in the manuscript.

Thank you once again for your constructive criticism. Your feedback has added value to the manuscript and has made it more readable.

Reviewer 3 Report

Comments and Suggestions for Authors

The authors have addressed most of the comments, however, additional detailed images need to be added in 400X in order to see clearly the osteoblast and osteoclast. The same for immunostaining.

Author Response

We appreciate your feedback, which has greatly improved the manuscript. Thank you for taking the time to review the article.

  • The authors have addressed most of the comments, however, additional detailed images need to be added in 400X in order to see clearly the osteoblast and osteoclast. The same for immunostaining.

The correction that you reviewed and approved has been applied.

Sincerely.

Round 3

Reviewer 2 Report

Comments and Suggestions for Authors

Not all concerns are followed, please:

Omit the mean+SD values in table 1 as the data were not normally distributed

Add the Q1 and Q3 values. When using min and max values just an outlier could be indicated. You can show the min and max values, but only when the IQR1 and IQR3 values are also shown

Author Response

The Kruskal-Wallis H test was utilized because the data didn't follow a normal distribution. The p value and the test used are stated in a footnote under the table. Using minimum and maximum instead of IQR provides more comprehensive information about the data. Presenting the mean, standard deviation, median, minimum, and maximum values in a table provides a comprehensive overview of the data. Commenting that the median was higher than the other group based on the median data interpretation would be incorrect. Reducing data crowding and increasing readability can be achieved by avoiding writing each group's median, minimum, and maximum values. It was deemed more appropriate to present the information in the table.